# A comprehensive comparison of tools for fitting mutational signatures

Matúš Medo [1,2] ✉, Charlotte K. Y. Ng [2,3] & Michaela Medová [1,2]

Mutational signatures connect characteristic mutational patterns in the genome with biological or chemical processes that take place in cancers. Analysis of mutational signatures can help elucidate tumor evolution, prognosis, and therapeutic strategies. Although tools for extracting mutational signatures de novo have been extensively benchmarked, a similar effort is lacking for tools that fit known mutational signatures to a given catalog of mutations. We fill this gap by comprehensively evaluating twelve signature fitting tools on synthetic mutational catalogs with empirically driven signature weights corresponding to eight cancer types. On average, SigProfilerSingleSample and SigProfilerAssignment/MuSiCal perform best for small and large numbers of mutations per sample, respectively. We further show that ad hoc constraining the list of reference signatures is likely to produce inferior results. Evaluation of real mutational catalogs suggests that the activity of signatures that are absent in the reference catalog poses considerable problems to all evaluated tools.

Since their introduction a decade ago[1,2], mutational signatures have become a widely used tool in genomics[3,4]. They allow researchers to move from individual mutations in the genome to biological or chemical processes that take place in living tissues[5,6]. The activity of various mutational signatures can also serve as prognostic or therapeutic biomarkers[7–9]. For example, homologous recombination deficiency leads to the accumulation of DNA damage and manifests itself in a specific mutational signature (signature SBS3 from the COSMIC catalog)[10,11]. Signature activities have been used to attribute mutations to endogenous, exogenous, and preventable mutational processes[12] and clock-like mutational signatures can help determine the absolute timing of mutations[13].

Mutational signatures can be introduced for single base substitutions (SBS), doublet base substitutions, small insertions and deletions[14], copy number alterations[15], structural variations[16], and RNA singlebase substitutions[17]. We focus here on SBS-based mutational signatures which are most commonly used in the literature. Current SBS signatures are defined using 6 possible classes of substitutions (C > A, C > G, C > T, T > A, T > C, and T > G) together with their two immediate neighboring bases, thus giving rise to $6 \times 4 \times 4 = 96$ different nucleotide contexts into which all SBS mutations in a given sample

are classified. De novo extraction of signatures from somatic mutations in sequenced samples has been used to gradually map the landscape of mutational signatures in cancers. Over time, the initial catalogue of 22 SBS-based mutational signatures in the first version of the Catalogue Of Somatic Mutations In Cancer (COSMIC) released in August 2013 has expanded to 86 signatures in the COSMICv3.4 version released in October 2023. This expansion was possible owing to the increased availability of whole exome sequencing (WES) and whole genome sequencing (WGS) data as well as improved tools for signature extraction. Extensive benchmarking of extraction tools on synthetic data has recently shown that SigProfilerExtractor outperforms other methods in terms of sensitivity, precision, and false discovery rate, particularly in cohorts with >20 active signatures[18]. A two-step process consisting of first extracting common signatures and then rare signatures has been recently recommended in ref. 19.

Nevertheless, the analysis of WGS and WES profiles of >23,000 cancers[18] has only discovered four new signatures and, in general, the likelihood of discovering new signatures in small studies is low. The more relevant task in most smaller studies is thus the fitting of existing signatures to given sequenced samples. In this process, the catalogs of somatic mutations are used to determine the signature contributions

[1]Department of Radiation Oncology, Inselspital, Bern University Hospital, University of Bern, Bern, Switzerland. [2]Department for BioMedical Research, Inselspital, Bern University Hospital, University of Bern, Bern, Switzerland. [3]IRCCS Humanitas Research Hospital, Rozzano, Milan, Italy. ✉e-mail: matus.medo@unibe.ch

to each individual sample. Many different tools have been developed for this task[4] (see Methods for their description and classification). However, while tools for extracting mutational signatures have been recently extensively benchmarked on synthetic data by various studies[14,18,20,21], a similar quantitative comparison is lacking for tools for fitting mutational signatures. This need is exacerbated by substantial variations between results obtained by different methods[22]. Furthermore, newly introduced tools for fitting mutational signatures are commonly compared with only a few existing tools, rarely the most recent ones, and a standardized comparison methodology is lacking. In this study, our aim is to fill this gap and provide a comprehensive evaluation of a broad range of fitting tools on synthetic data motivated by various types of cancer.

We constrain on fitting SBS signatures as they are the most widely used signature type. We generate two classes of synthetic mutational catalogs. In the first one, only one mutational signature is responsible for all mutations (single-signature cohorts). In the second one, signature activities in each sample are modeled after the activities found in real tumor samples from various cancers (heterogeneous cohorts). We have collected twelve tools for fitting mutational signatures, from earlier tools such as deconstructSigs to very recent ones such as MuSiCal. We assess their performance by comparing the known true signature activities in the synthetic catalogs with results obtained by various fitting tools. We find that there is no single fitting tool that performs best regardless of how many mutations are in the samples and which cancer type is chosen to model signature activities. Averaged across eight considered cancer types, SigProfilerSingleSample performs best when the number of mutations per sample is small (below 1000, roughly). For a higher number of mutations per sample,

SigProfilerAssignment and MuSiCal become the best performing tools. We also compare the tools by how prone they are to overfitting caused by increasing the size of the reference signature catalog and evaluate whether it is beneficial to constrain the reference catalog based on which signatures seem to be absent or little active in the analyzed samples. Real mutational catalogs are used to support the results obtained on synthetic catalogs. The analysis of real mutational catalogs further suggests that the activity of signatures that are absent in the reference catalogs challenges all evaluated fitting tools. We close with a discussion of open problems in fitting mutational signatures.

## Results

### Evaluation in single-signature cohorts

SBS-based mutational signatures are defined using the mutated base and its two neighboring bases. The total number of different "neighborhoods" (nucleotide contexts) to which each individual SBS can be attributed is 96 (6 different possible substitutions times four possible 5′ neighbors times four possible 3′ neighbors). This high number of contexts allows for a fine-grained classification of mutations and a detailed differentiation of many different mutational processes. On the other hand, it is a source of considerable sampling noise when the number of mutations is small. This is illustrated in Fig. 1a, b which shows the fraction of mutations in different contexts for two common signatures: Signature SBS1 with four distinct peaks among the C > T mutational contexts and signature SBS5 that lacks such peaks. While the peaks of SBS1 are clearly distinguished with as few as 100 mutations, the relative variations are much greater for SBS5 where the same number of mutations is effectively distributed among a larger number of contexts.

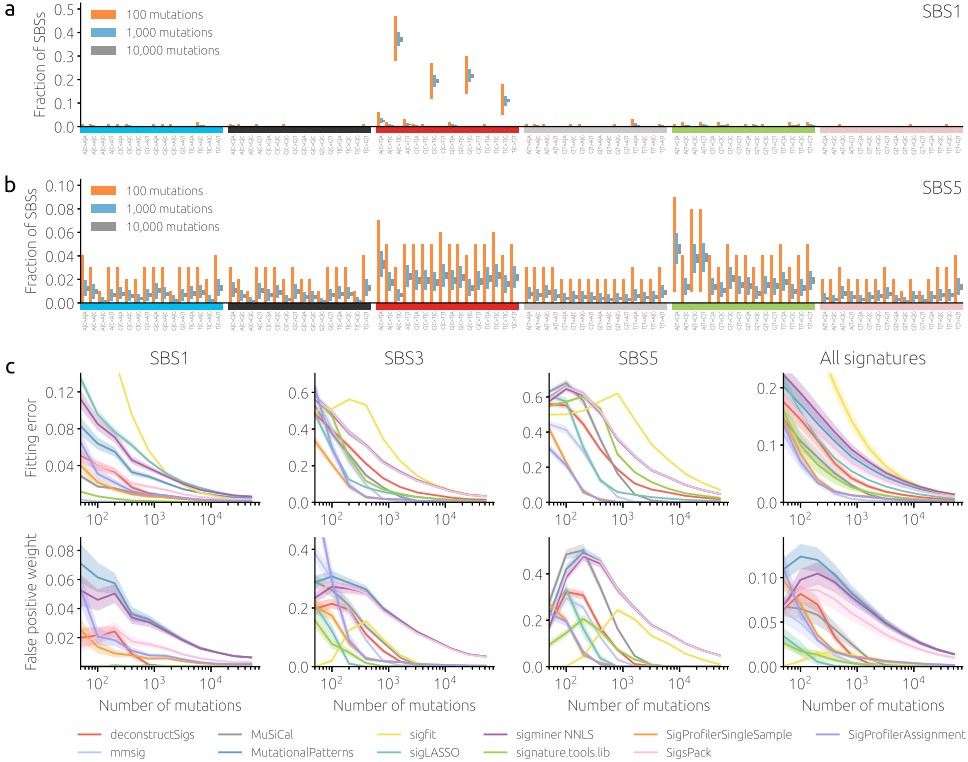

**Fig. 1 | The effect of the number of mutations in single-signature cohorts.** **a**, **b** The bars show the 95th percentile range of the observed fraction of mutations in synthetic data for SBS mutational contexts (horizontal axis) and different mutation counts (100, 1,000, and 10,000 represented with red, blue, and gray bars, respectively). Panels a and b show signature SBS1 with four distinct C > T peaks and signature SBS5 with a flat mutational spectrum, respectively. **c** Mean fitting error (top row) and mean total weight assigned to false positive signatures (bottom row) as functions of the number of mutations for three specific signatures (SBS 1, 3, and 5) and averaged over 49 non-artifact signatures from COSMICv3. Solid lines and shaded areas mark mean values and standard errors of the means, respectively, obtained from synthetic cohorts with 100 samples where all mutations are due to one signature. The catalog of all 67 COSMICv3 signatures was used for fitting by all tools.

**Table 1 | Basic information on the evaluated tools**

| Package | Language | URL |
|---|---|---|
| deconstructSigs | R | https://github.com/raerose01/deconstructSigs |
| mmsig | R | https://github.com/evenrus/mmsig |
| MuSiCal | Python | https://github.com/parklab/MuSiCal |
| MutationalPatterns | R | https://bioconductor.org/packages/release/bioc/html/MutationalPatterns.html |
| sigfit | R | https://github.com/kgori/sigfit |
| sigLASSO | R | https://github.com/gersteinlab/siglasso |
| sigminer | R | https://shixiangwang.github.io/sigminer |
| signature.tools.lib | R | https://github.com/Nik-Zainal-Group/signature.tools.lib |
| SigProfilerAssignment | Python | https://github.com/AlexandrovLab/SigProfilerAssignment |
| SigProfilerSingleSample | Python | https://github.com/AlexandrovLab/SigProfilerSingleSample |
| SigsPack | R | https://bioconductor.org/packages/release/bioc/html/SigsPack.html |
| YAPSA | R | https://bioconductor.org/packages/release/bioc/html/YAPSA.html |

We first evaluate the performance of signature fitting tools (see Table 1 for the list) in a scenario where all samples have 100% contributions of one given signature (single-signature cohorts). This scenario is clearly unrealistic: Depending on the type of cancer, the number of contributing signatures is 1–11 in individual samples and 5–22 in the cohorts available on the COSMIC website https://cancer.sanger.ac.uk/signatures/sbs/. Nevertheless, single-signature cohorts allow us to quantify substantial differences between the signatures as well as between signature-fitting tools. Figure 1c shows results for the "easy" signature SBS1 (a clock-like signature correlating with the age of the individual), "difficult" signatures SBS3 (related to DNA damage repair), SBS5 (present in virtually all samples), and the average over all non-artifact signatures in COSMICv3. The results are highly heterogeneous across signatures, fitting tools, and numbers of mutations: Six different tools achieve the lowest fitting error for at least one signature and number of mutations per sample (Supplementary Fig. 1). While some tools are among the best for SBS1 (signature.tools.lib and MuSiCal), they are among the worst for SBS5. The fitting error is approximately inversely proportional to the square root of the number of mutations for all fitting tools (Supplementary Fig. 2).

To understand the relation between the signature profile and their fitting difficulty, we compute their exponentiated Shannon index and find that it correlates highly (Spearman's rho 0.78–0.84, depending on the number of mutations) with the average fitting error achieved by the evaluated fitting tools (Supplementary Fig. 3). The correlation further improves (Spearman's rho 0.86–0.90) when the exponentiated Shannon index is multiplied with a measure of signature similarity with other signatures. We can conclude that while the fitting difficulty of a signature is mainly determined by the flatness of its profile (as measured by the Shannon index), its similarity to other signatures contributes as well.

The ranking of tools by the total weight assigned to false positives is substantially different, with sigLASSO, signature.tools.lib, and sigfit among the best performers when the number of mutations is small. Above 10,000 mutations per sample, false positives are largely avoided by all tools except for sigfit, MutationalPatterns, SigsPack, YAPSA, and sigminer. Notably, the running time of the evaluated tools spans over nearly four orders of magnitude (Supplementary Fig. 4) between SigsPack (the fastest method) and mmsig (the slowest method). For some tools (SigProfilerSingleSample, deconstructSigs, mmsig), the running time increases with the signature fitting difficulty represented by the fitting error (Supplementary Fig. 5). Nevertheless, even the longest running times of several minutes per sample are acceptable for common cohorts comprising tens or hundreds of samples; fitting mutational signatures is not a bottleneck in genomic data analysis[23].

## Evaluation in heterogeneous cohorts

We now move to synthetic datasets with empirical heterogeneous signature weights. Here, we use absolute signature contributions (i.e., the number of mutations attributed to a signature) in WGS-sequenced samples from various cancers as provided by the COSMIC website (see Methods and Supplementary Fig. 6). For further evaluation, we choose eight types of cancer with mutually distinct signature profiles (Supplementary Fig. 7): Hepatocellular Carcinoma (Liver-HCC), Stomach Adenocarcinoma (Stomach-AdenoCA), Head and Neck Squamous Cell Carcinoma (Head-SCC), Colorectal Carcinoma (ColoRect-AdenoCA), Lung Adenocarcinoma (Lung-AdenoCA), Cutaneous Melanoma (Skin-Melanoma), Non-Hodgkin Lymphoma (Lymph-BNHL), and Glioblastoma (CNS-GBM). The first four cancer types all have SBS5 as the main contributing signature but substantially differ in the subsequent signatures. The remaining four cancer types have different strongly contributing signatures: SBS4, SBS7a, SBS5 and SBS40, and SBS40, respectively. The relative signature weights in individual samples were then used to construct synthetic datasets with a given number of mutations, allowing us to assess the performance of the fitting tools in realistic settings. Compared with the previous scenario with single-signature samples, there are now two main differences. First, nearly all samples have more than one active signature (the highest number of active signatures in one sample is eleven). Second, signature contributions differ from one sample to another; the average cosine distance between signature contributions in different samples ranges from 0.19 for Liver-HCC to 0.50 for ColoRect-AdenoCA. This scenario is thus referred to as heterogeneous cohorts.

We evaluate the fitting tools on heterogeneous cohorts with different numbers of mutations (100, 2,000, and 50,000) that cover the common range of mutational burden in WES and WGS analyses. Heterogeneous cohorts are more difficult to fit than the previously single-signature cohorts. For 2,000 mutations, for example, the lowest mean fitting error is 0.055 (achieved by SigProfilerAssignment) whereas the fitting error of mmsig for the same number of mutations is below 0.01 for all signatures. This is a direct consequence of heterogeneous signature weights: Even when the number of mutations is as high as 10,000, a signature with a relative weight of 2% contributes only 200 mutations and, as we have seen, the fitting errors are high for such a small number of mutations. Overall, SigProfilerSingleSample has the lowest fitting error for 100 mutations per sample. SigProfilerAssignment becomes the best method, with mmsig close second, for 2,000 mutations per sample. SigProfilerAssignment and MuSiCal are the best methods by a large margin for 50,000 mutations per sample. The best-performing methods are similar when binary classification metrics (precision, sensitivity, and the F1 score) are considered (see Supplementary Fig. 8 for further evaluation metrics). Fitting methods based

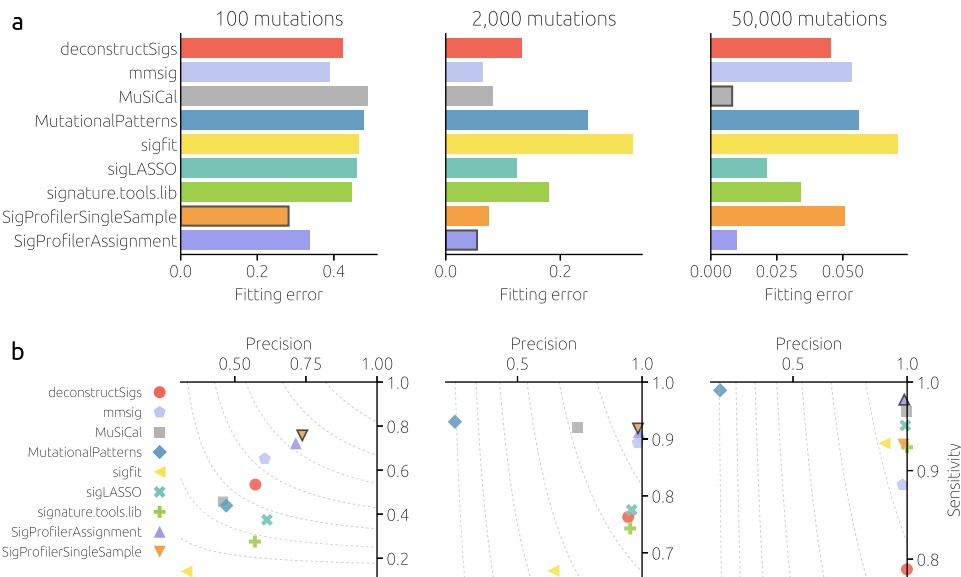

**Fig. 2 | A comparison of signature fitting methods on heterogeneous cohorts.** Mean fitting error (**a**) and precision and sensitivity (**b**) for different numbers of mutations per sample (columns) for the evaluated fitting tools. The results are averaged over 50 cohorts with 100 samples for eight distinct cancer types (see Methods). Each tool used all 67 COSMICv3 signatures as the reference catalog. The best-performing tool in each panel is marked with a frame (in the bottom row, the best tool by the F1 score combining precision and sensitivity; the dashed contours correspond to F1 = 0.9, 0.8, … ). Results are not shown for SigsPack, YAPSA, and all three variants of sigminer as they are close (fitting error correlation above 0.999) to the results of MutationalPatterns.

on nonnegative least squares (represented by MutationalPatterns in Fig. 2) suffer from low precision due to overfitting. Flat signatures such as SBS5 and SBS40 are commonly underrepresented by the fitting tools (Supplementary Fig. 9) and the tools disagree on them most (Supplementary Fig. 10).

Tool performance differs greatly by cancer type (Supplementary Fig. 11). For 2,000 mutations, three different tools achieve the lowest fitting error for individual cancer types (SigProfilerAssignment for four of them, mmsig for three, and SigProfilerSingleSample for one). For 50,000 mutations, MuSiCal and SigProfilerAssignment are the best methods for five and three cancer types, respectively (see Supplementary Fig. 12 for the dependence on the number of mutations). The results are similar when other performance metrics are considered (Supplementary Fig. 13). The overall similarity between the fitting error values (Supplementary Fig. 14) shows two distinct clusters of tools: the first formed by signature.tools.lib and all tools directly based on nonnegative least squares and the second including deconstructSigs, sigLASSO, and MuSiCal.

One of the best-performing tools, SigProfilerSingleSample, reports a sample reconstruction similarity score that is often used to remove the samples whose reconstruction score is low. Our results show that this score is strongly influenced by the number of mutations; its absolute value is thus a poor indicator of the fitting accuracy (Supplementary Fig. 15). The same is the case of SigProfilerAssignment (Supplementary Fig. 16).

**Choosing the reference catalog**

The chosen reference catalog of signatures can significantly impact the performance of fitting algorithms[4,24]. When the newer COSMICv3.2 catalog of mutational signatures is used as a reference instead of COSMICv3, the fitting error increases for most methods (Supplementary Fig. 17) due to an increased number of signatures (from 67 to 78) which makes the methods more prone to overfitting. However, SigProfilerSingleSample, SigProfilerAssignment, mmsig, MuSiCal, and sigLASSO are robust to increasing the number of reference signatures when the number of mutations per sample is high. On the other hand, when the reference signatures are constrained to the signatures that have been previously observed for a given cancer type and to artifact

signatures[4], the fitting error decreases substantially for most methods (Supplementary Fig. 18). Methods that are robust to adding reference signatures benefit less from removing irrelevant signatures, which in turn diminishes differences between methods (Supplementary Fig. 19).

To better illustrate the effect of reducing the number of reference signatures, we use three selected methods (SigsPack which together with MutationalPatterns and sigminer is the most sensitive to the reference catalog, SigProfilerSingleSample, and sigLASSO) and classify their output to true positive signatures, false positive signatures that are relevant to a given cancer type and false positive signatures that are irrelevant to a given cancer type (Fig. 3a). Using the whole COSMICv3 as a reference (top row), a simple method (SigsPack) starts with 17% of relevant false positives and 56% of irrelevant false positives. For comparison, these numbers are only 21% and 27% for SigProfilerSingleSample and 8% and 23% for sigLASSO (which, furthermore, leaves 50% of the mutations unassigned). When only relevant signatures are used as reference (bottom row), SigsPack improves much more than the two other methods. This further demonstrates how simple methods are particularly sensitive to the reference catalog and overfitting. Another interesting observation is that while, regardless of the reference catalog, SigsPack and sigLASSO tend to zero false positives as the number of mutations increases, this is not the case for SigProfilerSingleSample (as shown in Supplementary Fig. 11, SigProfilerSingleSample performs poorly for Stomach-AdenoCA used in Fig. 3a). SigProfilerSingleSample evidently has a powerful algorithm to infer the active signatures from few mutations, but it does not converge to true signature weights when the mutations are many.

Nevertheless, relying on a pre-determined list of relevant signatures is problematic for a number of reasons. First, the lists of signatures active in specific cancers are likely to change over time. Four of the eight considered cancer types have cohorts with <100 patients, so adding more WGS-sequenced tissues to the catalog is likely to significantly expand the list of signatures that are active in them. Second, most tools have difficulty recognizing that the provided signature catalog is insufficient even when the number of mutations in a sample is very large (Supplementary Fig. 20; we return to this observation below). The estimated signature activities can therefore change in the future when better analytical tools become available. Third, when the

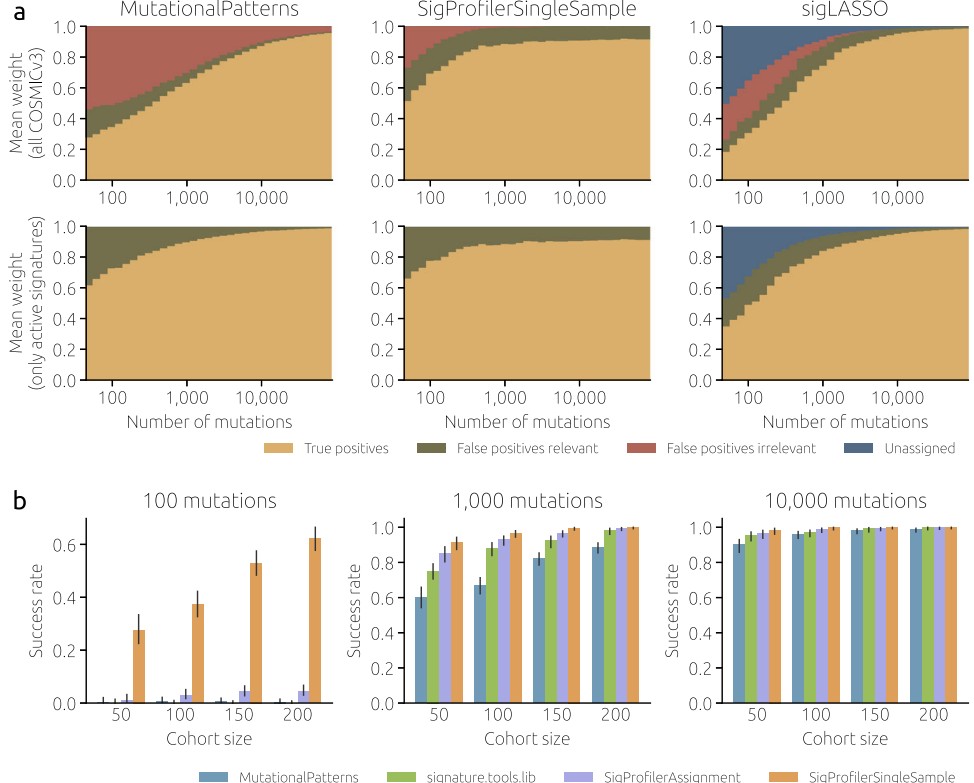

**Fig. 3 | Practical consequences of increasing the mutational burden. a** As the number of mutations per sample increases, weights assigned to false positives decrease (relevant false positives: excess weights assigned to signatures active in the cohort; irrelevant false positives: weights assigned to signatures not active in the cohort). We used simulated input data with 100 samples and Stomach-AdenoCA signature weights. The reference signature catalog is COSMICv3 (top row) and the 18 signatures active in the Stomach-AdenoCA samples (bottom row). **b** Performance of four fitting methods in identifying systematic differences in mutational weights between two groups of samples (see Methods for details). The success rate is the fraction of 250 synthetic cohorts with artificially introduced differences in SBS40 weights between even- and odd-numbered samples where the estimated SBS40 weights differ significantly (Wilcoxon rank-sum test, *p*-value below 0.05). The error bars show the 95% confidence interval (Wilson score interval).

list of relevant signatures is obtained from the COSMIC website, we rely on results obtained with one specific tool and this tool can be biased. We thus employ a different approach, which is based on fitting signatures using the whole COSMICv3 catalog (step 1) and then constraining the reference catalog to the signatures that sufficiently occur in the obtained results (step 2, see Methods for details).

The two-step fitting process can substantially improve the performance of some methods in some cases (Supplementary Fig. 21). At the same time, it tends to be detrimental to well-performing methods when the number of mutations per sample is high. It is then better to refrain from an ad hoc procedure for choosing a subset of reference signatures and instead rely on statistical selection mechanisms built into the methods themselves. A different multi-step process to select reference signatures was proposed in ref. 3. Signatures are first extracted de novo, each extracted signature is then assigned to one or two known reference signatures (see Methods for details), and thus-identified reference signatures are then used for fitting. This approach can also improve the fitting performance (Supplementary Fig. 22). For samples with 100 mutations, the best results in terms of the fitting error and F1 score are obtained using SigProfilerSingleSample and SigProfilerAssignment, respectively, combined with the former two-step process to trim the reference signatures. For samples with 2,000 or 50,000 mutations, it is best to use the complete COSMIC catalog as a reference.

### Signature fitting for downstream analysis
We have focused so far on estimates of signature weights and their errors with respect to a well-defined ground truth. In many cases,

however, the estimates are only important as input for further downstream analyses assessing the correlations of signature weights with some clinicopathological parameters. We present here a simplified example of such an analysis by creating synthetic cohorts with CNS-GBM signature weights where statistically significant differences in the weights of signature SBS40 between even- and odd-numbered samples are introduced (see Methods for details). Estimation errors, together with the actual magnitude of the effect and the cohort size, are crucial to the ability to detect a significant difference in the signature weights. We contrast four fitting tools: simple MutationalPatterns, SigProfilerSingleSample and SigProfilerAssignment which perform well in Fig. 2, and widely used signature.tools.lib. When the number of mutations is large (10,000 in Fig. 3b), all tools are sufficiently precise to identify a statistically significant difference between the two groups of samples in nearly all cohorts. By contrast, when the number of mutations is smaller, the choice of the fitting tool matters. At 1000 mutations per sample, SigProfilerSingleSample is successfull in >90% of cohorts, while the other tools perform worse, in particular when the cohort size is small. At 100 mutations per sample, SigProfilerSingleSample still has some statistical power to detect a difference in SBS40 activities between the two groups of samples whereas the other tools fail regardless of the cohort size. When the signature of interest is easier to fit than SBS40 used in Fig. 3b, the differences between fitting tools become smaller (Supplementary Fig. 23).

### Evaluation on real mutational catalogs
To assess the performance of signature fitting tools on real data, we use mutational catalogs of real WGS samples made available by the

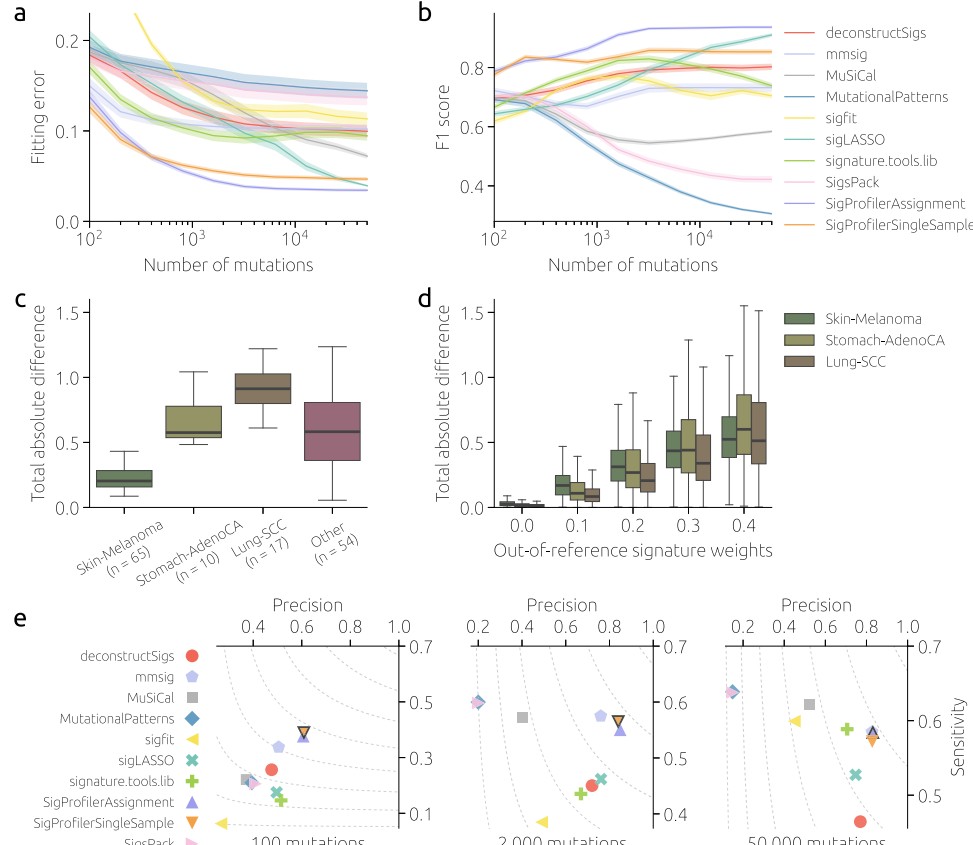

**Fig. 4 | A comparison of signature fitting methods on real mutational catalogs and adverse synthetic catalogs. a, b** Fitting error and F1 score obtained on four chosen PCAWG mutational catalogs subsampled to varying numbers of mutations. Performance metrics are computed using the ground truth computed on full-size mutational catalogs. Solid lines and shaded areas mark mean values and their 95% confidence intervals, respectively, obtained from 200 independent catalog sub-samplings for each number of mutations. **c** Differences between signature estimates computed by SigProfilerAssignment, sigLASSO, and MuSiCal for the 146 PCAWG mutational catalogs with >50,000 mutations. Boxes show the quartiles of the data and whiskers indicate the extent of the data up to 1.5 × IQR (same for **d**). **d** Differences between signature estimates computed by SigProfilerAssignment, sigLASSO, and MuSiCal for synthetic mutational catalogs with 50,000 mutations per sample and increasing activity of signatures that are absent in COSMICv3 (horizontal axis; see Methods for details). **e** Performance of signature fitting tools for synthetic heterogeneous cohorts where two signatures absent from COSMICv3 have both weights 10%. The results are averaged over 50 cohorts with 100 samples for eight distinct cancer types (see Methods). The tool with the highest F1 score is highlighted in each panel. Results are not shown for YAPSA, and all three variants of sigminer as they are close to the results of MutationalPatterns and SigsPack. COSMICv3 was used as a reference by the evaluated tools in all panels.

International Cancer Genome Consortium (ICGC). In particular, we select four PCAWG samples that all have >50,000 mutations, their mutational catalogs differ, and well-performing fitting tools agree best in their signature estimates (see Methods for details). The average signature weights estimated by the three tools that perform best for 50,000 mutations in Fig. 2, SigProfilerAssignment, MuSiCal, and sigLASSO, are used as the ground truth for these data. Results obtained on subsampled mutational catalogs then allow us to measure the performance of signature fitting tools as a function of the number of mutations (Fig. 4a, b). We see again that SigProfilerSingleSample is the best tool when the number of mutations is small. SigProfilerAssignment becomes the best tool when the number of mutations exceeds 1,000; sigLASSO's performance becomes competitive for the largest mutational burdens.

Although the fitting tools that we used to define the ground truth produce similar results for the four chosen samples, the general level of disagreement for all ICGC catalogs with >50,000 mutations is high (Fig. 4c). This contradicts the results shown in Fig. 2 where the low fitting errors achieved by the best-performing tools for 50,000 mutations imply that their respective estimated signature weights must be close. This is confirmed by the left-most point in Fig. 4d which corresponds to the heterogeneous cohorts shown in Fig. 2. However, when signatures that are absent in the reference catalog COSMICv3 are introduced in synthetic catalogs (see Methods for details), the signature estimates produced by the fitting tools show a level of disagreement that grows with the contribution of the out-of-reference signatures (Fig. 4d). This demonstrates that the 67 COSMICv3 signatures cannot be used to compensate the contributions of signatures that are absent from the reference catalog. Although the situation where some active signatures are absent in the reference catalog should be avoided, it is important to assess how different tools cope with this adverse setting. Compared to Fig. 2b, fitting performance worsens considerably when out-of-reference signatures contribute 20% of all mutations in synthetic catalogs (Fig. 4e). Not only sensitivity decreases due to out-of-reference signatures whose activity cannot be identified by construction but precision also decreases as the fitting fools struggle to "redistribute" the out-of-reference mutations to other COSMICv3 signatures. In summary, these results show that using an incomplete reference catalog is even more problematic than over-fitting due to an extensive reference catalog.

We finally study correlations between clinical parameters and signature activities estimated in real data by different fitting tools. In particular, we consider platinum signatures in primary and recurrent ovarian cancer samples (Supplementary Fig. 24) and signatures associated with defective DNA mismatch repair in PCAWG samples (Supplementary Fig. 25). The obtained results again document overfitting

of tools such as MutationalPatterns and SigsPack. In addition, MuSiCal, mmsig, and sigLASSO (in this order) also show elevated levels of false positive rates.

## Discussion

The broad range of tools for fitting mutational signatures makes it difficult to understand which tool to choose for a given project. In this work, we provide a comprehensive assessment of twelve different tools and their variants on synthetic and real mutational catalogs. Using mutational catalogs where only one signature is active allows us to quantify the differences in the fitting difficulty of individual signatures. We find that flat signatures whose average similarity with other signatures is high are the most difficult to fit. To assess the fitting tools, we create synthetic mutational catalogs whose signature activities are modeled after eight distinct cancer types. We use cohorts with 100 mutations per sample (common for WES), 2,000 mutations per sample, and 50,000 mutations per sample (common for WGS). We find that when the number of mutations is small (100 mutations per sample), SigProfilerSingleSample is the best tool to use for all cancer types. As the number of mutations increases, SigProfilerAssignment and MuSiCal become the best tools while mmsig is best for some cancer types for an intermediate number of mutations (2,000 mutations per sample). The results are similar when other evaluation metrics are used instead of fitting error for the evaluation (Supplementary Fig. 8). Results obtained using real mutational catalogs (Fig. 4) further support the conclusions made using synthetic data.

The risk of overfitting the data by including too many signatures in the reference catalog is often discussed in the literature[3,4,19]. We show that two methods (SigProfilerSingleSample and SigProfilerAssignment) are robust to such overfitting when the number of mutations per sample is 2,000 or more and several other methods are robust for 50,000 mutations per sample (Supplementary Fig. 17). Crucially, the common practice of excluding signatures from the reference catalog because they seem to be little active in the analyzed data is often little effective or even harmful. We find that it is beneficial only for 100 mutations per sample in combination with SigProfilerSingleSample or SigProfilerAssignment. In other cases it is preferable to use a well-performing fitting tool and a complete COSMIC catalog as a reference.

While our work gives clear recommendations on which tools to consider and which to avoid, many issues can be addressed in the future to further improve the fitting of mutational signatures. First, a similar assessment of fitting tools can be done for other types of signatures: double base substitutions, small insertions and deletions, copy number variations, and structural variations. Then, some tools (e.g., sigfit and mmsig) can compute confidence intervals for their estimates. For sigfit, we used these estimates in the way recommended by the authors: When the lower bound of the confidence interval for the relative signature weight is below 0.01, the signature is marked as absent in the sample (its relative weight is set to zero). Our results show that this recommended practice indeed improves the results achieved by sigfit. By bootstrapping (resampling the original mutational catalogs with replacement), confidence intervals could be computed also for the tools that do not compute them by themselves. It would be worth investigating whether thus-determined confidence intervals could also improve the performance of other fitting tools.

To fit mutational signatures, it is commonly required that each sample has at least 50[25] or 200[26] single base substitutions. However, different signatures are differently difficult to fit (Fig. 1), so it is unlikely that a universal threshold is meaningful. Similarly, the common practice of requiring that sample reconstruction accuracy exceeds 0.90 has little support in our results (Supplementary Figs. 15 and 16). Based on the simulation framework established here, it would be possible to study the minimum necessary number of mutations for different signatures of interest and different cancer types. To best match the needs

of practitioners, it would be possible to extend the simulation framework so that it finds the best-performing fitting tool for a given cancer type, a list of signatures of interest, and a given distribution of sample mutation counts.

When an active signature is missing from the reference catalog because it is unknown or falsely deemed inactive in a given cohort, most fitting tools cannot cope with this situation and distribute all (or nearly all) mutations among signatures from the reference set (Supplementary Fig. 20). Three tools—deconstructSigs, signature.tools.lib, and sigfit—are less prone to this issue but they are not among the best performers in our other evaluations. Analysis of real mutational catalogs with high mutational burden shows that the tools that perform well on synthetic data produce widely divergent signature estimates for many samples. The same behavior is reproduced using synthetic data with 50,000 mutations when a substantial fraction of all mutations (20–40%) is due to signatures that are absent in the reference catalog that is used for fitting. Besides increasing disagreement between the estimates obtained by different tools, we show that out-of-reference signatures substantially lower the precision and sensitivity of the fitting results. Taken together, our results suggest that underfitting due to incomplete reference catalogs poses a bigger challenge to the estimation of signature activities than often-discussed overfitting due to extensive reference catalogs, in particular when the sample mutational burden is high.

To alleviate the detrimental effects of out-of-reference signatures, one could quantify whether estimated signature activities are likely to produce a given mutational catalog (ensemble approaches to combine results from various tools[27] could be first used to reduce the variability of results shown in Fig. 4c). A sample that fails such a check possibly features signatures that are absent in the reference catalog. One could then identify the mutations that are compatible with signatures from the reference catalog. The remaining mutations can be left unassigned or, based on results from other samples in the analyzed cohort, assigned to novel signatures. A similar concept of common and rare mutational signatures has been recently introduced in the context of signature extraction de novo[19].

More generally, COSMIC signatures are the result of analyzing tumor samples from many different organs, so they can be viewed as an average across them[19]. The fast-growing number of sequenced samples makes it possible to construct organ-specific reference catalogs that better reflect mutational processes that occur in these biologically diverse systems.

## Methods

### Reference signatures

We used all signatures from version 3.0 of the COSMIC catalog (COSMICv3, https://cancer.sanger.ac.uk/signatures/sbs/), in particular the SBS signatures for human genome assembly GRCh38. COSMICv3 contains 67 mutational signatures (18 are artifact signatures) defined in 96 mutational contexts. While newer COSMIC versions (the latest version is 3.4 from October 2023) added more signatures, profiles of the signatures present in COSMICv3 have not been altered (all absolute weight differences are below $10^{-7}$).

To measure the distinctiveness of signature profiles, we compute their exponentiated Shannon index which is a common measure of diversity[28] that can be understood as the effective number of active nucleotide contexts in a mutational signature. For a signature whose weight is concentrated in one context, the exponentiated Shannon index is one. For a signature with the same weight 1/96 in all contexts, the exponentiated Shannon index is 96. For COSMICv3 signatures, the exponentiated Shannon index ranges from 2.7 (SBS48) to 80.3 (SBS3).

### Synthetic mutational catalogs

Absolute contributions of signatures to sequenced tissues from various cancer types were downloaded from the Catalogue Of Somatic

Mutations In Cancer (COSMIC, https://cancer.sanger.ac.uk/signatures/sbs/, catalog version 3.2) on November 14, 2021 for all 46 non-artifact SBS signatures that have tissue contribution data available. These data included only samples with reconstruction accuracy above 0.90 and for each sample, all signatures that contribute at least 10 mutations. Using the provided unique sample identifiers, we compiled the relative signature contributions in individual WGS-sequenced samples that are stratified by the cancer type. We chose eight different cancer types for further analysis: Hepatocellular carcinoma (Liver-HCC, $n = 422$), stomach adenocarcinoma (Stomach-AdenoCA, $n = 170$), head and neck squamous cell carcinoma (Head-SCC, $n = 57$), colorectal carcinoma (ColoRect-AdenoCA, $n = 72$), lung adenocarcinoma (Lung-AdenoCA, $n = 62$), cutaneous melanoma (Skin-Melanoma, $n = 280$), non-Hodgkin Lymphoma (Lymph-BNHL, $n = 117$), and glioblastoma (CNS-GBM, $n = 63$). All signatures in the tissue contribution data are from the COSMICv3 catalog.

We created a cohort of 100 samples with $m$ mutations for each cancer type as follows. We first chose 100 samples (with repetition) from all samples for which empirical signature weights were available. Denoting the relative weights of signature $s$ for sample $i$ (sample composition) as $w_{si}$ and the relative weight of context $c$ for signature $s$ (signature profile) as $\omega_{cs}$, the relative weight of context $c$ for sample $i$ is obtained as a weighted sum over all signatures,

$$W_{ci} = \sum_s \omega_{cs} w_{si}. \qquad (1)$$

We then generated a synthetic mutational catalog with these weights by distributing $m$ mutations among the 96 contexts, while the probability that a mutation is assigned to context $c$ in sample $i$ is $W_{ci}$. This is mathematically equivalent to first choosing signature $s$ for sample $i$ with probability $w_{si}$ and then choosing context $c$ with probability $\omega_{cs}$. As a result, the number of mutations in contexts are multinomially distributed with mean $mW_{ci}$ for context $c$ and sampe $i$. The mean number of mutations contributed to sample $i$ by signature $s$ is $mw_{si}$. When this number is smaller than 10, the signature cannot be correctly identified by the evaluated tools as we use the standard procedure of setting the weights of signatures that contribute less than ten mutations to zero. To not bias the evaluation, we: (1) For sample $i$ and $m$ mutations per sample, remove all signatures that have $mw_{si} < 10$ and (2) normalize the weights of the remaining signatures to one.

The approach described above allows us to reproduce empirical signature weights in previously analyzed samples without resorting to assumptions such as a log-normal distribution of the number of mutations due to a given signature[14,18] or adding additional zeros to the Poisson distribution to reproduce signatures that are not active in a sample[20]. The code for generating synthetic mutational catalogs and evaluating the signature fitting tools, *SigFitTest*, is available at https://github.com/8medom/SigFitTest.

For Fig. 3b, we created synthetic mutational catalogs with signature activities modeled after real CNS-GBM samples where systematic differences in the activity of signature SBS40 have been introduced. After generating sample weights as described in the previous paragraph, the relative weights of SBS40 were multiplied by 1.3 and divided by 1.3 for even- and odd-numbered samples, respectively. The relative weights of all signatures in each sample were then normalized to one. Signature SBS40 is active in nearly all CNS-GBM samples (Supplementary Fig. 7) and its median relative weight is close to 50%. By the described process, the median weights in the two groups of samples become 42% and 55%, respectively. In another analysis (Supplementary Fig. 23), we introduced differences in the weights of SBS1 for CNS-GBM samples by multiplying and dividing it by 1.2 in the two respective groups of samples.

To generate samples with signatures that are absent in the reference catalog COSMICv3, we used the 12 signatures that have been

added in COSMICv3.3.1 (SBS10c, SBS10d, SBS86, SBS87, SBS88, SBS89, SBS90, SBS91, SBS92, SBS93, SBS94, SBS95). For a sample with the total weight of out-of-reference signatures $x$, we first multiply the weights of all COSMICv3 signatures by $1 - x$. We then choose two signatures from the list above at random and assign them weights $0.7x$ and $0.3x$ (for Fig. 4d) or $0.5x$ each (Fig. 4e where $x$ is fixed to 0.2, i.e., out-of-reference signatures contribute 20% of all mutations in each sample).

## Tools for signature fitting

The task of fitting known mutational signatures to a mutational catalog consists of finding the combination of signatures from a reference set that "best" matches the given catalog. Denoting the matrix with reference signatures as R, where $R_{cs}$ is the relative weight of context $c$ for signature $s$, and the given normalized mutational catalog as m, where $m_{ci}$ is the fraction of mutations in context $c$ in sample $i$, this amounts to solving

$$m = Rw \qquad (2)$$

with respect to w, where $w_{si}$ is the relative weight of signature $s$ in sample $i$. To allow for a unique solution, vectors representing weights of different signatures must be linearly independent. The number of reference signatures thus cannot be higher than the number of mutational contexts (96 contexts for the common SBS signatures). Equation (2) is thus an over-determined system of linear equations. Several fitting tools are therefore based on minimizing the difference between $m$ and R$w$ through non-negative least squares (as the signature weights cannot be negative) or quadratic programming. We evaluated several tools that belong to this class: MutationalPatterns v3.14.0[26], YAPSA v1.30.0[29], SigsPack v1.18.0[30], and sigminer v2.3.1[31] which has three separate methods based on quadratic programming, non-linear least squares, and simulated annealing. We find that all these tools produce similar results.

Other tools use various iterative processes by which the provided set of reference signatures is gradually reduced by removing the signatures, for example, whose inclusion does not considerably improve the match between the observed and reconstructed mutational catalogs (or, opposite, signatures are gradually added as long as the reconstruction accuracy sufficiently improves). We evaluated deconstructSigs v1.9.0[32], SigProfilerSingleSample v0.0.0.27[14], SigProfilerAssignment v0.1.7[33], mmsig v0.0.0.9000[24], and signature.tools.lib v2.2.0[19], that all belong to this category. The newest tool, SigProfilerAssignment, combines backward and forward iterative adjustment of the reference catalog and these steps are repeated until convergence.

Finally, sigLASSO v1.1 combines the data likelihood in a generative model with L1 regularization and prior knowledge[34]. To allow for a fair comparison with other tools, we did not use prior knowledge when evaluating sigLASSO. In our experience, this tool sometimes fails to halt but starting it again with the same input data resolves the issue. A similar Bayesian framework is used by sigfit v2.2.0[35]. MuSiCal v1.0.0 is a recent tool which uses likelihood-based sparse NNLS to fit signatures[36].

While most tools produce signature weights that sum to one when normalized by the number of mutations, signature.tools.lib and sigLASSO explicitly allow for unassigned mutations. We used standard parameter settings for all tools. The authors of sigfit recommend setting all signatures whose lower bounds of the estimated 95% confidence intervals are below 0.01 to zero. The authors of deconstructSigs recommend setting all signatures whose relative weights are below 0.06 to zero. All results shown for sigfit and deconstructSigs follow these recommendations. Similarly to the COSMIC database, activities of signatures that contribute <10 mutations are set to zero. YAPSA makes it possible to use signature-specific cut-offs. However, the tool has only one set of precomputed cut-offs whose derivation is not clearly documented and these cut-offs work

poorly for some cancer types. We thus do not use signature-specific cut-offs for YAPSA.

## Evaluation metrics

Fitting error quantifies the difference between the true relative signature weights (which are used to generate the input mutational catalogs) and the estimated relative signature weights (for tools that estimate absolute signature weights, those are first normalized by the number of mutations in each sample). Denoting the true and estimated relative weights of signature $s$ in sample $i$ as $w_{si}$ and $\tilde{w}_{si}$, respectively, the absolute error is summed over all signatures,

$$\sum_s |w_{si} - \tilde{w}_{si}|/2, \tag{3}$$

and then averaged over all samples. The division by two is introduced for normalization purposes. The lowest fitting error, 0, is achieved when the estimated signature weights are exact for all signatures and all samples. The highest fitting error, 1, is achieved when the estimated signature weights sum to one for each sample but they are all false positives. For example, when a sample has 40% contribution of SBS1 and 60% of SBS4 but a tool estimates 20% contribution of SBS2 and 80% contribution of SBS13, the fitting error is 1. We further quantify the agreement between the true and estimated signature weights by computing their Pearson correlation. For each sample where at least three signatures have positive either true or estimated weights, we compute the Pearson correlation coefficient between these two vectors whilst excluding all signatures that are zero for both of them. The obtained values are averaged over all samples in the cohort.

Further common evaluation metrics are based on classifying the estimated signatures as true positives (when both $\tilde{w}_{si}$ and $w_{si}$ are positive), false positives (when $\tilde{w}_{si} > 0$ and $w_{si} = 0$), true negatives (when $\tilde{w}_{si} = 0$ and $w_{si} = 0$), and false negatives (when $\tilde{w}_{si} = 0$ and $w_{si} > 0$)[18,20,33]. False positive weight quantifies how much weight is assigned to the signatures that are not active in the corresponding samples. The value for sample $i$,

$$\sum_{s:w_{si}=0} \tilde{w}_{si}, \tag{4}$$

is averaged over all samples. The lower the value, the better. Precision quantifies the reliability of the identified active signatures. It is computed by averaging the number of true positives to all positives over all samples in the cohort. Sensitivity quantifies the ability to identify all active signatures. It is computed by averaging the number of true positives to the number of active signatures (i.e., the sum of true positives and the false negatives) over all samples. Higher sensitivity can be commonly achieved at the cost of lower precision[37]. This is addressed by the F1 score which is the harmonic mean of precision and recall (we compute the F1 score for each sample and then average it over all samples). To achieve a good F1 score, high precision and sensitivity are necessary.

## Selecting reference signatures

It has been argued that the removal of irrelevant reference signatures can improve the fitting results[5]. To assess this hypothesis, we test a two-step process where: (1) We fit the samples using all COSMICv3 signatures as a reference and (2) keep only the signatures whose relative weight exceeds threshold $w_0$ for at least 5 samples in our cohorts with 100 samples. We use thresholds $w_0 = 0.1$ ($m = 100$), $w_0 = 0.03$ ($m = 2000$) and $w_0 = 0.01$ ($m = 50,000$) which correspond to absolute signature contributions 10, 60, and 500, respectively. This pruning of reference signatures is often beneficial (Supplementary Fig. 21). However, for a high number of mutations and well-performing

methods (mmsig, MuSiCal, sigLASSO, SigProfilerAssignment, SigProfilerSingleSample), this two-step process increases the produced fitting errors ($m = 50,000$ in Supplementary Fig. 21). When lower thresholds are used to keep signatures, this can be avoided but there is nevertheless no improvement for $m = 50,000$ and the improvements for $m = 100$ vanish (results not shown).

We also assess the approach based on the extraction of signatures de novo as suggested in ref. 3. For signature extraction, we use SignatureAnalyzed v0.0.7 (https://github.com/getzlab/SignatureAnalyzer) with L1 prior on both W and H and a Poisson objective function for optimization. As recommended in ref. 3, each extracted signature is then compared with individual signatures from COSMICv3 as well as linear combinations of two signatures from COSMICv3. The signature (or a pair of signatures) that yields the smallest cosine distance is then added to the list of trimmed reference signatures for the given input data. When the number of mutations is small ($m = 100$), less than five signatures are selected, on average. When the number of mutations is large ($m = 50,000$), 90% of the active signatures are selected, on average. This approach can improve the fitting results (Supplementary Fig. 22), in particular when the number of mutations is small. However, a direct comparison of the results obtained by the two described approaches shows that the former process based solely on signature fitting produces the lowest fitting error/highest F1 score for $m = 100$ when combined with SigProfilerSingleSample and SigProfilerAssignment, respectively. For more mutations, it is best to use SigProfilerAssignment, mmsig, or MuSical and use all COSMICv3 signatures as a reference.

The recent tool MuSiCal[36] has a so-called full pipeline where de novo signature discovery is followed by matching the extracted signatures to a given catalog and refitting the data using the matched catalog signatures. The matching procedure is principled and involves automatic optimization of model parameters. At the same time, MuSiCal's full pipeline is prohibitively computationally demanding for large-scale evaluation in many different settings as we do here.

We finally note that any approach to select a subset of reference signatures based on the input mutational catalogs is affected by the cohort size. For a small cohort, de novo signature extraction is likely to produce inferior results which will affect the subsequent refitting. All results shown here apply to cohorts of 100 samples.

## Real mutational catalogs

To complement the synthetic data, we used real mutational catalogs produced by the International Cancer Genome Consortium (ICGC). Their PCAWG datasets[14] are available from the ICGC upon request. From 2,780 WGS PCAWG samples, we kept the 146 samples that had at least 50,000 single base substitutions. Selecting the samples with high mutational burden is motivated by Fig. 2 where small fitting errors are achieved for 50,000 mutations. Estimates of signature activities in these samples by SigProfilerAssignment, sigLASSO, and MuSiCal (that perform best in Fig. 2 for 50,000 mutations) were used to choose the samples where the three tools disagree the least. To quantify the disagreement, we used the total absolute difference in estimated signature weights between two methods, averaged over all three pairs of methods. To prevent choosing samples that are overly similar to each other, we required that the total absolute difference between the mutational profiles must be at least 0.5. The input mutational profiles as well as the reconstructed profiles corresponding to the estimated signature weights agree well for the four chosen samples (Supplementary Fig. 26). The ground truth signature estimates were obtained by averaging the positive signature weights over the three fitting tools; signatures that have been found active by less than two tools are set to zero. Thus-obtained signature weights were then normalized to one. Active signatures in sample 1 are SBS2 (50%), SBS13 (49%), and SBS1 (1%). Active signatures in sample 2 are SBS7a (88%) and SBS7b (12%). Active signatures in sample 3 are SBS5 (37%), SBS2 (36%), SBS13 (26%),

and SBS1 (1%). Active signatures in sample 4 are SBS7a (58%), SBS7b (30%), SBS7c (7%), and SBS7d (5%). To study correlations of signature estimtes with clinical parameters, we used the clinical data of the OV-AU project contained in the ICGC Data Portal data Release 28 (2019-11-26) and the mismatch repair status of PCAWG samples (proficient/deficient) provided by ref. 36 in Supplementary Table 4.

## Reporting summary
Further information on research design is available in the Nature Portfolio Reporting Summary linked to this article.

## Data availability
The absolute contributions of signatures to sequenced tissues from various cancers were obtained from the Catalogue Of Somatic Mutations In Cancer (COSMIC), https://cancer.sanger.ac.uk/signatures/sbs/. They are also included in SigFitTest (https://github.com/8medom/SigFitTest). The reference catalogs of single base substitution (SBS) signatures are available from COSMIC, https://cancer.sanger.ac.uk/signatures/downloads/. The synthetic datasets that were used to evaluate the fitting tools can be generated by function generate_synthetic_catalogs() of SigFitTest. The mutational catalogs of 146 WGS PCAWG samples with at least 50,000 mutations, obtained from the ICGC, as well as the consensus ground truth based on sigLASSO, SigProfilerAssignment, and MuSiCal for four samples where the three tools agree best, are included in SigFitTest.

## Code availability
The code of SigFitTest is available at https://github.com/8medom/SigFitTest[38]. Links to the code of the evaluated fitting tools are provided in Table 1.

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

## Acknowledgements
We thank Peter M. Degen, Yitzhak Zimmer, Daniel M. Aebersold, and Kjong-Van Lehmann for their valuable support.

## Author contributions

Ma.M., C.N. and Mi.M. designed the study. Ma.M. developed the simulation code and analyzed the results. Ma.M., C.N. and Mi.M. discussed the results and contributed to the final manuscript.

## Competing interests

The authors declare no competing interests.
