## [Transparent Peer Review file · Nature Communications]

A comprehensive comparison of tools for fitting mutational signatures

Corresponding Author: Dr Matus Medo

Version 0:

Reviewer comments:

Reviewer #1

(Remarks to the Author)

This manuscript, authored by Matuš Medo and Michaela Medová, under the title "A Comprehensive Comparison of Tools for Fitting Mutational Signatures," delves into a crucial aspect of mutational signature assignment in cancer genomic studies. The authors have conducted meticulous benchmarks for mutational signature fitting, providing essential signature activity data for downstream analyses, including associations with genomic features, clinical variables, and exposure data in tumor etiology studies. The thorough evaluation of 11 signature fitting tools using synthetic input data across various scenarios showcases the manuscript's depth, covering elements such as cancer types, mutation counts, signature flattening, reference signatures, and single/heterogeneous cohorts.

The manuscript is well-written, employs a clear methodology and is poised to make a significant contribution to advancing the mutational signature field. However, I have three suggestions to further enhance this manuscript:

1) While the authors commendably benchmarked and recommended tools based on accuracy and performance, it would be valuable to explore and discuss how signature fitting and tools might differ when considering interactions between two or more distinct signatures (e.g., two signatures from the same etiology, such as SBS2 + SBS13). Additionally, for heterogeneous cohorts, how consistent is the fitting among different tools? Discrepancies in signature assignment tools are common (in most cases, one tool assigns a correct number of mutations to one signature, and another tool assigns 0 mutations to the same signatures), and It will be important to understand the factors influencing these variations, which would aid in selecting the most appropriate assignment tools.

2) Sharing the synthetic data used in the study is crucial for facilitating comparisons with new tools developed by others. Providing access to this data would enable researchers to assess the performance of their tools against a standardized set, fostering reproducibility and furthering the field.

3) Additionally, The author listed the tools used in the study. I would also recommend including their respective versions and download dates used in this study. Given the continuous updates to these tools, this information is essential for readers to contextualize the findings and consider potential advancements or changes in tool functionalities over time.

Reviewer #2

(Remarks to the Author)

This study investigates and compares the performance of 11 fitting tools for mutational signatures using in-silico data. The authors highlight the differences and advantages of specific tools and caution against the use of a pre-selected catalogue of signatures, particularly in samples with low mutational burden (100 mutations). While the paper is well-written and addresses an important topic related to the standardization and reproducibility of mutational signatures analysis, it has notable limitations.

Despite its utility, reliance solely on in-silico data is suboptimal given the availability of datasets and studies with over 10,000 patients. Without validation using real data, the true validity of the authors' claims cannot be adequately assessed. In-silico data are known to introduce biases that may lead to overperformance of certain fitting tools, particularly when dealing with a plethora of signatures, such as the 80+ SBS signatures.

One critical aspect of mutational signature analysis is not merely quantification but accurately identifying which signatures are present or absent. The use of in-silico data introduces a potential bias, leading to an overestimation of the performance of certain fitting tools. It is essential for the authors to investigate whether specific signatures, such as the platinum signature, are exclusively identified in patients with a history of platinum exposure or if they also manifest as false-positive signals in other cases. A similar evaluation should be conducted for signatures like SBS7 UV light and skin cancers. The challenge is magnified with an extensive set of signatures, exceeding 80 SBS signatures, where fitting all signatures may result in an increased number of false positives. Addressing these issues will enhance the robustness of the mutational signature analysis and contribute to more accurate and reliable findings.

Many of the recent tools in mutational signature analysis operate based on cosine similarity. It's noteworthy that the emphasis is not solely on the total number of mutations but rather on the distinctiveness of the 96-mutational profile. This crucial consideration is often overlooked in in-silico-only experiments. Recognizing the importance of distinct mutational profiles, beyond sheer mutation quantity, is essential for a more nuanced and accurate interpretation of mutational signatures.

Fitting an in-silico signature with 100% contribution is suboptimal for several reasons. Firstly, in-silico analyses typically lack the level of noise present in real samples. Secondly, it is established that SBS1 and SBS5 are universally present in all human cells. A more accurate experimental approach would involve generating in-silico data using SBS1, SBS5, and a third signature of choice. The subsequent analysis should then be adjusted to account for the increased contribution of the third signature. This methodology better mirrors the complexity and noise inherent in real-world genomic data, leading to more reliable and applicable findings.

The authors should also delve into the concept of tissue-specific signatures recently reported by Degasperis et al. (Science 2022). The notion that a common signature can exhibit slight variations across different tissues underscores another reason why relying solely on in-silico data is suboptimal.

The critiques against the 2-level analysis are deemed acceptable but may lack sufficient robustness. For the first point: while acknowledging that mutational signatures can change, with only two COSMIC versions in the last 10 years, we can consider them as reasonably stable and consistent. For the second point: the fitting issues are attributed not to the size of the mutational burden but to the challenges in detecting distinct, well-defined profiles against flat signatures or background. Approaches that include all signatures have been historically criticized for generating numerous mistakes and errors. To fully embrace this approach, more than in-silico results are deemed necessary. The authors should have conducted their analysis on datasets such as PCAWG to evaluate biological consistency. For instance, examining the exclusive presence of UV light signatures in skin cancer, SBS4 in tobacco smoking-related cancers, or platinum in relapsed patients exposed to chemotherapy would provide a more comprehensive and convincing validation.

Reviewer #3

(Remarks to the Author)

Medo and Medova carried out a benchmark of eleven mutational signature assignment tools with synthetic data to objectively identify how each tool behaves at different mutation burdens. The use of a large set of synthetic tumours makes the findings of the study robust and is an important piece of work for the community. There are key insights to the set of reference signatures used and provide practical advice for researchers working on the topic. The authors have carried out substantial work to compare state-of-the-art signature assignment tools but there are several comments to be addressed to improve the quality of the manuscript.

A major comment is that the main figures can be improved by incorporating the supplementary figures into the main figures to support the findings or include more supporting figures. For example, supplementary Figure 4 or a version of the figure can be incorporated into the main figures. Figure 2 can benefit from a summary table of specificity and sensitivity scores to quantitatively show the performance of each tool at each mutational burden. Figure 3 can be extended to show the examples of false positive and false negative assignments and common issues including highly similar signatures which can be misassigned.

Several key insights to the assignment of mutational signatures are lacking in the study. Figure 1 demonstrates the fitting error profiles of each tool with only 1 signature present. The authors have mentioned in the manuscript that the testing scenario is clearly unrealistic but has little to add other than the fact that flat signatures are hard to assign (SBS3, SBS5 and SBS40) and different tools have drastically different error profiles at different mutation burdens. There is a pattern of peaks in Figure 1C with each panel with 2 tools performing poorly throughout and they should be mentioned in the main text. Similarly, the panel with SBS12 and SBS18 shows a high error profile and the authors can comment on what is likely influencing the error profiles. In addition, figure 1C can also benefit from a color bar placed at the bottom to denote the tool tested in each column so that the reader does not need to use the positions of the bars to identify which tool is represented.

The stratification of the synthetic tumours based on mutation burden can seem abstract and hard to appreciate. Can the authors comment on the number of tumours analysed in PCAWG in each of the 8 cancer types stratified by mutation burden? Could the authors comment on the variation of assignments in the real patient tumours and how this will affect interpretation of the results? This analysis is crucial to provide context in patient tumours and demonstrate the importance of selecting assignment tools to have reliable signature assignment calls.

The authors have provided an extensive assessment of the performance of tools and including accuracy and runtime. The findings are practical and important for the selection of signature assignment tools. Could the authors test ensemble tools such as EnsembleFit <https://academic.oup.com/bib/article/24/6/bbad331/7280728#418642340> and if an ensemble approach is comparable to the best performing tool identified in this study.

A minor comment that would improve the discussion on reference signature sets would be to test organ specific signatures reference set by Degasperis et al 2022 (<https://signal.mutationalsignatures.com/>) for the 8 cancer types to determine the changes in errors with organ specific signatures. The set of common and rare signatures can be discussed as well but there are no studies showing how useful the approach is.

Last, the authors used a set of averaged evaluation metrics including fitting error, false positive weights. This approach limits the granularity of understanding possible explanations of why the error rates are different or which are the neighbourhoods contributing to this change. Is it possible to use error rates that are averaged not throughout 96 dimensions but in the 6 mutation classes (C>A, C>T ... etc) or per trinucleotide to identify if specific mutational classes are driving the error rate changes? This would be useful as a heatmap on the heterogenous cohort to identify what are the signatures that are commonly false positive or negative assigned by tools. This will help inform software users on the common misassignments and perhaps signatures to check during their analyses more carefully.

Version 1:

Reviewer comments:

Reviewer #1

(Remarks to the Author)

I have reviewed the revisions and the author's responses to my comments regarding the manuscript. The author has thoroughly addressed all the issues I raised, providing detailed explanations and supplementary data where needed. The additions of empirical data on signature correlations and the variations in signature assignment by different tools greatly enhance the clarity and depth of the study. Furthermore, the commitment to sharing synthetic data and the inclusion of specific tool versions used in the study are valuable for ensuring reproducibility and facilitating future research. Therefore, I have no further comments and believe the manuscript has been significantly improved.

Reviewer #2

(Remarks to the Author)

The Authors have adequately addressed most of my comments, and I believe the paper in its current version presents a valuable study for the community.

The addition of an analysis on PCAWG patients is a significant improvement. Currently, it is limited to samples with a hypermutated profile, typically attributed to a single process (such as SBS2, APOBEC, MSI, etc.). While this is useful, it might be beneficial to extend this analysis to samples with lower and more admixed mutational signatures. For example, among PCAWG WGS samples exposed to platinum-based treatments (specifically SBS31), how accurately do different tools predict the presence and quantity of platinum signatures? While I don't believe this analysis is strictly necessary, validating it could be useful because platinum is a distinct signature detectable only in the presence of exposure to the drug.

Reviewer #3

(Remarks to the Author)

The authors have addressed my comments and the additional analysis on cancer tumours have proved to be insightful. Thank you for the insightful paper.

A comprehensive comparison of tools for fitting mutational signatures

Response to the reviewers

Matúš Medo, Charlotte K.Y. Ng, Michaela Medová

June 6, 2024

Summary of changes

We thank the editor for handling our manuscript and reviewers for their comments which helped us to improve it. The main changes to the manuscript are:

1. We evaluate one more fitting method, MuSiCal (Nature Genetics, 2024).
2. Figures 1 and 2 have been merged.
3. Figure 2 (previously Figure 3) has been extended with binary classification metrics (precision, sensitivity, and F1 score). Figure S8 in Supplementary Information (SI) shows further evaluation metrics.
4. Figure 3 combines previous figures 4 and 6.
5. Figure 5 has been removed and the discussion that followed the original figure 5 was shortened to save space for new results.
6. Evaluation on real data has been added (Figure 4) which both strengthens the previous results obtained on synthetic data as well as puts more emphasis on the problem of an incomplete reference catalog where some signatures active in the analyzed samples are missing.

We elaborate on these changes in our point-by-point response to the reviewers below. Substantial modifications are highlighted in the revised manuscript.

Reviewer #1

We thank the reviewer for a positive overall evaluation of our work.

1) While the authors commendably benchmarked and recommended tools based on accuracy and performance, it would be valuable to explore and discuss how signature fitting and tools might differ when considering interactions between two or more distinct signatures (e.g., two signatures from the same etiology, such as SBS2 + SBS13).

These correlations are indeed important. By using empirical signature weights in WGS-sequenced tissues from various cancers as provided by the COSMIC website, these correlations are naturally reproduced in our synthetic mutational catalogs. For example, signatures SBS2 and SBS13 indeed occur together; they are particularly active in Head-SCC and Lung-AdenoCA data (see SI, Fig. S7). In the empirical Head-SCC contribution data, the Pearson correlation of weights of SBS2 and SBS13 is as high as 0.96 and this high correlation is then reproduced (up to sampling variations) in the generated synthetic catalogs. In summary, by relying on empirical signature weights on real samples, we create realistic signature compositions without making complicated assumptions about signature activities and their interactions.

Additionally, for heterogeneous cohorts, how consistent is the fitting among different tools? Discrepancies in signature assignment tools are common (in most cases, one tool assigns a correct number of mutations to one signature, and another tool assigns 0 mutations to the same signatures), and it will be important to understand the factors influencing these variations, which would aid in selecting the most appropriate assignment tools.

This is an interesting question; we have added figures in SI which address it. Fig. S9 shows the signatures that are most systematically under- or over-assigned by individual tools. It shows that flat signatures SBS5 and SBS40 are common “collection” signatures for mutations that do not fit elsewhere. Further, Fig. S10 shows the signatures on which the fitting tools disagree most. SBS5 and SBS40 feature prominently also here, but some signatures (e.g., SBS31) are actually very distinct, yet the weights assigned to them by different tools differ considerably.

Supplementary Figure 9: Largest average differences between estimated and true signature weights for individual tools and signatures in heterogeneous cohorts (a complement to Figure 2 in the main text).

Supplementary Figure 10: Signatures with the largest disagreement between the results obtained by different tools in heterogeneous cohorts (a complement to Figure 2 in the main text). The disagreement between tools is computed as the standard deviation of the weights estimated by different tools for a given sample, averaged over 50 cohorts with 100 samples for eight different cancer types.

2) Sharing the synthetic data used in the study is crucial for facilitating comparisons with new tools developed by others. Providing access to this data would enable researchers to assess the performance of their tools against a standardized set, fostering reproducibility and furthering the field.

We agree that a shared collection of datasets for future testing and benchmarking is an important contribution. The code to reproduce most of the results in the paper is available online (<https://github.com/8medom/SigFitTest>). Function `generate_synthetic_catalogs()` can be used to generate the synthetic datasets used in our paper. We hope that this code will help establish a common testing platform for fitting mutational signatures.

3) Additionally, the author listed the tools used in the study. I would also recommend including their respective versions and download dates used in this study. Given the continuous updates to these tools, this information is essential for readers to contextualize the findings and consider potential advancements or changes in tool functionalities over time.

Thank you, tool versions can be found in the Methods section.

1 Reviewer #2

We thank the reviewer for the assessment of our work and critical yet constructive remarks.

This study investigates and compares the performance of 11 fitting tools for mutational signatures using in-silico data. The authors highlight the differences and advantages of specific tools and caution against the use of a pre-selected catalogue of signatures, particularly in samples with low mutational burden (100 mutations). While the paper is well-written and addresses an important topic related to the standardization and reproducibility of mutational signatures analysis, it has notable limitations.

Thank you for this summary of our work and for the appreciation of its importance. We address specific comments below.

Despite its utility, reliance solely on in-silico data is suboptimal given the availability of datasets and studies with over 10,000 patients. Without validation using real data, the true validity of the authors' claims cannot be adequately assessed. In-silico data are known to introduce biases that may lead to overperformance of certain fitting tools, particularly when dealing with a plethora of signatures, such as the 80+ SBS signatures.

In-silico validation indeed has its pitfalls and it is important both question the model used to generate the synthetic data as well as consider alternatives (use of real data). Our answer to this point is thus two-fold. First, we attempted to set up a highly realistic model with empirically-driven signature contributions. The model makes few assumptions and we hope that it can become a reference point for future works on signature estimation. Second, we have now added an evaluation of the fitting tools on real mutational catalogs. A whole new figure (Figure 4 in the revised manuscript) has been introduced to present the respective results. This is an important addition as it both supports the results obtained previously on synthetic mutational catalogs and highlights the importance of signatures that are absent in the reference catalog (this aspect is also addressed by Fig. S20 in SI). We show that out-of-reference signatures increase the disagreement between fitting results obtained by different tools. Thank you again for this suggestion.

One critical aspect of mutational signature analysis is not merely quantification but accurately identifying which signatures are present or absent. The use of in-silico data introduces a potential bias, leading to an overestimation of the performance of certain fitting tools. It is essential for the authors to investigate whether specific signatures, such as the platinum signature, are exclusively identified in patients with a history of platinum exposure or if they also manifest as false-positive signals in other cases. A similar evaluation should be conducted for signatures like SBS7 UV light and skin cancers. The challenge is magnified with an extensive set of signatures, exceeding 80 SBS signatures, where fitting all signatures may result in an increased number of false positives. Addressing these issues will enhance the robustness of the mutational signature analysis and contribute to more accurate and reliable findings.

The problem with verifying the etiology of signatures is very topical [see, for example, <https://doi.org/10.1038/s41588-020-0692-4> where the authors show that “most agents (carcinogens with chronic exposure) do not generate distinct mutational signatures”]. We view our benchmarking paper as a stepping stone for further analyses that can address the correlations mentioned by you. To be able to do that well, several challenges identified in our work need to be addressed, in particular the problem of signatures that are absent from the reference catalog. When such signatures are active in the analyzed samples, they induce disagreement between the

signature estimates obtained by different tools and increase the fitting errors. How to recognize problematic samples (and, in the second step, fit them better) is an open question.

Many of the recent tools in mutational signature analysis operate based on cosine similarity. It's noteworthy that the emphasis is not solely on the total number of mutations but rather on the distinctiveness of the 96-mutational profile. This crucial consideration is often overlooked in in-silico-only experiments. Recognizing the importance of distinct mutational profiles, beyond sheer mutation quantity, is essential for a more nuanced and accurate interpretation of mutational signatures.

This is a valid point and the main reason why we have devoted substantial space (Figure 1) to the discussion of how the number of mutations *and* the signature profile determine the fitting error. Supplementary Figure 3 shows that the Shannon entropy of signature profiles strongly correlates with the fitting error and this correlation further increases when also the mean correlation with other signatures is considered. To make this point clear, we have expanded its discussion in the main text.

Fitting an in-silico signature with 100% contribution is suboptimal for several reasons. Firstly, in-silico analyses typically lack the level of noise present in real samples. Secondly, it is established that SBS1 and SBS5 are universally present in all human cells. A more accurate experimental approach would involve generating in-silico data using SBS1, SBS5, and a third signature of choice. The subsequent analysis should then be adjusted to account for the increased contribution of the third signature. This methodology better mirrors the complexity and noise inherent in real-world genomic data, leading to more reliable and applicable findings.

Generating in-silico mutational catalogs is facilitated by the linear activity of mutational signatures which is biologically plausible and mathematically simple. The noise is introduced naturally through random sampling and the relative weight of this noise is more pronounced for samples with few mutations, which makes them more difficult to fit. For Figure 4, we introduce another level of "noise" by assigning some activity to signatures that are absent from the reference catalog that is used for fitting.

Our heterogeneous cohorts go in the direction of realistic signature contributions that you described above. In particular, we use empirical signature contributions that are available from the COSMIC catalog and use them to generate synthetic data. In this way, not only heterogeneous signature activity in samples (SBS1 and SBS5 active in most samples, complemented by a varied set of other signatures depending on the considered cancer type) but also correlations between signatures (for our Head-SCC samples, Pearson correlation between SBS2 and SBS3 is above 0.9) are reproduced.

We believe that the model to create synthetic catalogs (publicly available at <https://github.com/8medom/SigFitTest>) is one of the main contributions of our work. We hope that it will help establish a common testing platform for fitting mutational signatures.

The authors should also delve into the concept of tissue-specific signatures recently reported by Degasperi et al. (Science 2022). The notion that a common signature can exhibit slight variations across different tissues underscores another reason why relying solely on in-silico data is suboptimal.

Yes, the rationale for using tissue-specific signatures is mentioned in Discussion. At the same time, the relative ranking of tools for fitting mutational signatures is unlikely to change substantially upon slight variations to the reference catalog. Our main conclusions on which tool to choose and

how to proceed with the analysis, how robust are different tools to overfitting, and how sensitive they are to incomplete reference catalogs, are thus still relevant.

The critiques against the 2-level analysis are deemed acceptable but may lack sufficient robustness. For the first point: while acknowledging that mutational signatures can change, with only two COSMIC versions in the last 10 years, we can consider them as reasonably stable and consistent.

We have not been sufficiently clear here. We have not meant changes to the reference catalog but changes to which signatures have been found active for specific cancer types. Tissue contribution data available from the COSMIC catalog include only dozens of WGS samples for some cancer types (e.g., 57 samples for Head-SCC). Upon analyzing more samples, the list of signatures active in Head-SCC samples can thus considerably grow. Also, the methods to estimate signature activity are still developing, so the previously estimated signature activities can change. But we agree that when the landscape of mutational activities will be mapped in detail, this method of constraining the reference catalog will become viable. We have made changes in the text to make our point more clear.

For the second point: the fitting issues are attributed not to the size of the mutational burden but to the challenges in detecting distinct, well-defined profiles against flat signatures or background. Approaches that include all signatures have been historically criticized for generating numerous mistakes and errors. To fully embrace this approach, more than in-silico results are deemed necessary. The authors should have conducted their analysis on datasets such as PCAWG to evaluate biological consistency. For instance, examining the exclusive presence of UV light signatures in skin cancer, SBS4 in tobacco smoking-related cancers, or platinum in relapsed patients exposed to chemotherapy would provide a more comprehensive and convincing validation.

We examined the sensitivity of fitting tools to overfitting (Fig. S17 in SI) and found that when the number of mutations is sufficiently high, several tools are robust to adding irrelevant signature to the reference catalog (e.g., the fitting errors of SigProfilerSingleSample and SigProfilerAssignment do not grow upon adding more signatures for 2000+ mutations per sample). Based on your another comment above, we have extensively evaluated the opposite situation where some relevant signatures are omitted from the reference catalog. Notably, reconstruction metrics that are computed by some of the well-performing tools do not identify the samples where the fitting error is high (Figs. S15 and S16 in SI). The problem of an incomplete reference catalog is clearly relevant to real samples where it is not clear if the observed mutational catalogs are compatible with the current reference catalogs. We find that no currently available fitting tool can deal with this problem satisfactorily. Once this issue is addressed, it will be the right time for a large-scale analysis of real mutational catalogs and correlations between signature activities with various clinical parameters. Our manuscript answers the first question on this path by identifying the tools that perform well in various situations.

2 Reviewer #3

We thank you for the assessment of our work and for the comments.

A major comment is that the main figures can be improved by incorporating the supplementary figures into the main figures to support the findings or include more supporting figures. For example, supplementary Figure 4 or a version of the figure can be incorporated into the main figures. Figure 2 can benefit from a summary table of specificity and sensitivity scores to quantitatively show the performance of each tool at each mutational burden. Figure 3 can be extended to show the examples of false positive and false negative assignments and common issues including highly similar signatures which can be misassigned.

This is a good point, thank you. We have now modified figures in the main text to make them more comprehensive. We have also extended the set of evaluation metrics to include binary evaluation metrics (shown in Figures 2 and 4 in the main text and Figs. S8 and S19 in SI). We decided not to promote Fig. SF4 (now Fig. SF1) as it concerns the very specific scenario of single-signature samples/cohorts. Figs. S9 and S10 in SI further show signatures that are systematically under- or over-assigned by specific tools and signatures on which the evaluated tools disagree most, respectively.

Several key insights to the assignment of mutational signatures are lacking in the study. Figure 1 demonstrates the fitting error profiles of each tool with only 1 signature present. The authors have mentioned in the manuscript that the testing scenario clearly unrealistic but has little to add other than the fact that flat signatures are hard to assign (SBS3, SBS5 and SBS40) and different tools have drastically different error profiles at different mutation burdens. There is a pattern of peaks in Figure 1C with each panel with 2 tools performing poorly throughout and they should be mentioned in the main text. Similarly, the panel with SBS12 and SBS18 shows a high error profile and the authors can comment on what is likely influencing the error profiles. In addition, figure 1C can also benefit from a color bar placed at the bottom to denote the tool tested in each column so that the reader does not need to use the positions of the bars to identify which tool is represented.

The heatmap in panel 1C was not the best way to compare the tools and signatures—we have now replaced it with the former Figure 2. This makes the figures more informative in line with your previous suggestion. The dependence between signature distinctiveness and the fitting error is specifically examined in Fig. S3 in SI. With respect to signatures SBS12 and SBS18, they are not flat but they also lack distinctive peaks which correctly puts them between flat signatures as SBS5, for example, and very distinct signatures as SBS17a, for example.

The stratification of the synthetic tumours based on mutation burden can seem abstract and hard to appreciate. Can the authors comment on the number of tumours analysed in PCAWG in each of the 8 cancer types stratified by mutation burden?

This is a good point. We have added a comment saying that 100 and 50,000 mutations are typical values for WES and WGS, respectively.

Could the authors comment on the variation of assignments in the real patient tumours and how this will affect interpretation of the results? This analysis is crucial to provide context in patient tumours and demonstrate the importance of selecting assignment tools to have reliable signature assignment calls.

A similar comment has been raised by Reviewer #2. We have now added evaluation of the fitting tools on real mutational catalogs (Fig. 4a,b in the main text) and find a good agreement with our observations made on synthetic mutational catalogs. Analysis of real mutational catalogs further highlights the importance of signatures that are absent in the reference catalog (Fig. 4c in the main text; this aspect is also addressed by Fig. S20 in SI). We show that out-of-reference signatures can increase the disagreement between fitting results obtained by different tools (Fig. 4d). While out-of-reference signatures worsen the performance of all fitting tools, the best-performing tools change little (Fig. 4e as compared to Fig. 2b).

Analysis of real mutational catalogs and the subsequent analysis of synthetic catalogs with out-of-reference signatures are important changes that have enriched our manuscript. Thank you again for this suggestion.

The authors have provided an extensive assessment of the performance of tools and including accuracy and runtime. The findings are practical and important for the selection of signature assignment tools. Could the authors test ensemble tools such as EnsembleFit <https://academic.oup.com/bib/article/24/6/bbad331/7280728> and if an ensemble approach is comparable to the best performing tool identified in this study.

This is an interesting question. In Figure 2, performance differences between various tools are so large (see Figure 2) that combining their results cannot outperform individual best-performing methods. In particular, tools from the same family of methods as MutationalPatterns tend suffer from overfitting, false positives, and consequently low precision for any number of mutations per sample. Methods such as sigLASSO, signature.tools.lib, MuSiCal and sigfit need a large number of mutations per sample to perform well. Finally, when the number of mutations is high, tools such as SigProfilerAssignment and MuSiCal produce nearly perfect results which do not need to be improved further. (In fact, as we show in Figure 3b, all tools then produce signature estimates that can be similarly well used in downstream analysis.) When out-of-reference signatures are active in the analyzed samples, the suggested ensemble approach alone cannot solve this issue as all individual tools that could be combined suffer from out-of-reference signatures (according to our results, no current tool can deal with this problem well). However, it is possible that an ensemble approach can be useful as a part of the solution. We now mention this possibility in Discussion.

A minor comment that would improve the discussion on reference signature sets would be to test organ specific signatures reference set by Degasperi et al 2022 (<https://signal.mutationalsignatures.com/>) for the 8 cancer types to determine the changes in errors with organ specific signatures. The set of common and rare signatures can be discussed as well but there are no studies showing how useful the approach is.

Thank you for the suggestion. We have expanded the discussion of this reference in relation to the analysis of real mutational catalogs mentioned above.

Last, the authors used a set of averaged evaluation metrics including fitting error, false positive weights. This approach limits the granularity of understanding possible explanations of why the error rates are different or which are the neighbourhoods contributing to this change. Is it possible to use error rates that are averaged not throughout 96 dimensions but in the 6 mutation classes (C_iA , C_iT ... etc) or per trinucleotide to identify if specific mutational classes are driving the error rate changes? This would be useful as a heatmap on the heterogenous cohort to identify what are the signatures that are commonly false positive or negative assigned by tools. This will help inform software users on the common misassignments and perhaps signatures to check during their analyses more carefully.

Yes, this is a useful practical question. As mentioned above, Figs. S9 and S10 in SI show signatures that are systematically under- or over-assigned by specific tools and signatures on which the evaluated tools disagree most, respectively.

A comprehensive comparison of tools for fitting mutational signatures

Response to the reviewers

Matúš Medo, Charlotte K.Y. Ng, Michaela Medová

August 22, 2024

Summary of changes

We thank the editor for handling our manuscript and the reviewers for their comments which helped us to improve it. The main changes to the manuscript are:

1. Following the suggestions of reviewers 1 and 2, we improved the accompanying code available at <https://github.com/8medom/SigFitTest>.
2. Following the suggestion of reviewer 2, we evaluated correlations between: (1) estimated activity of platinum signatures and exposure to platinum-based drugs in ovarian cancer samples and (2) estimated activity of signatures associated with defective DNA mismatch repair (MMR) and the MMR status of PCAWG samples.

Changes in addition to the reviewers' comments:

1. We updated the evaluated fitting tools to the newest possible version.
2. We added confidence intervals in Figure 3b and Supplementary Figure 23 (they have been omitted erroneously before).

We provide a point-by-point response to the reviewers below. Modifications are highlighted in the revised manuscript.

Reviewer #1

We thank the reviewer for a positive overall evaluation of the revised manuscript.

I have reviewed the provided code and found that while it functions well with the included dataset, it requires significant modifications to be applicable to other datasets. I recommend that the authors revise the code to enhance its usability, making it more flexible for users who wish to apply it to different datasets. This would greatly increase the utility and accessibility of the code for a broader audience.

Thank you for this useful suggestion. We have now implemented an additional function in the code, `fit_external()`, which fits a provided mutational catalog and evaluates the estimated signature activities by comparing them with given ground truth activities.

Reviewer #2

We thank the reviewer for a positive overall evaluation of the revised manuscript.

The addition of an analysis on PCAWG patients is a significant improvement. Currently, it is limited to samples with a hypermutated profile, typically attributed to a single process (such as SBS2, APOBEC, MSI, etc.). While this is useful, it might be beneficial to extend this analysis to samples with lower and more admixed mutational signatures.

The necessity to only use samples with high mutational burden (in our case, 50,000 mutations or more) is one of the limitations of the used approach to evaluate fitting methods on real samples. While we agree that it constraints the diversity of the analyzed samples, there are two practical reasons that led us to this choice. First, our previous simulations on synthetic data show that error metrics (fitting error, precision, sensitivity) improve with the number of mutations (Figure 3). Using hypermutated samples thus allows us to define robust ground truth solutions by averaging the estimates obtained by well-performing methods. Second, the subsampling of mutational catalogs with many mutations allows us to inspect a broad range of mutational burdens (100–50,000 mutations) in Figure 4a. We have clarified the reason for this choice in the manuscript.

For example, among PCAWG WGS samples exposed to platinum-based treatments (specifically SBS31), how accurately do different tools predict the presence and quantity of platinum signatures? While I don't believe this analysis is strictly necessary, validating it could be useful because platinum is a distinct signature detectable only in the presence of exposure to the drug.

We found platinum exposure data for one specific PCAWG project (OV-AU). We analyzed 52 untreated primary samples and 10 recurrent samples treated with surgery and chemotherapy. As platinum-based anticancer drugs have been the standard chemotherapy for a long time, the recurrent samples can be presumed to be exposed to platinum. As expected, the platinum signatures SBS31 and SBS35 are frequently found in the primary samples by the overfitting-prone tools MutationalPatterns and SigsPack; MuSiCal and mmsig find them in two and one samples, respectively. By contrast, platinum signatures are common in the recurrent samples. The corresponding results are summarized in the new Supplementary Figure 24 (see below).

Supplementary Figure 24: Mutational signature analysis of untreated primary tumor ($n = 52$) and treated recurrent tumor ($n = 10$) samples from the OV-AU cohort for which SBS mutational catalogs are available. The recurrent samples have been treated with surgery and chemotherapy. As platinum-based anti-cancer drugs are the standard chemotherapy, the recurrent samples are presumed to be exposed to platinum. The fraction of samples where platinum signatures (SBS31 and SBS35) are active (left panel) and their mean joint activity in the samples where they are active (right panel). MutationalPatterns and SigsPack find a low activity of platinum signatures in 60% of the primary samples although these samples were not treated with platinum-based drugs. Most tools find platinum signatures in the majority of the recurrent samples. SigProfilerAssignment (30%) and SigProfilerSingleSample (50%) are more conservative in this respect and only attribute these signatures to the samples where their activity is high (right panel).

Additionally, we did a similar analysis based on the characterization of PCAWG WGS samples as mismatch repair deficient (MMRD, $n = 41$) or proficient (MMRP, $n = 2,700$) in “Accurate and sensitive mutational signature analysis with MuSiCal” (Nature Genetics, 2024). We used seven signatures that are associated with defective DNA mismatch repair (SBS6, SBS14, SBS15, SBS20, SBS21, SBS26, and SBS44) and evaluated fitting tools by how well their signature activity estimates can classify samples as MMR deficient or proficient (see the figure below, new Supplementary Figure 25). This classification task turns out to be relatively easy as all methods achieve high values of precision, sensitivity, and F1 score. While the relative ranking of the methods shares several similarities with our previous comparisons (MutationalPatterns, SigsPack, and deconstructSigs perform worst while SigProfilerAssignment is among the best performers), sigfit, and signature.tools.lib perform better compared to Figure 3 and Figure 4. However, this is only one specific test of the correlations between clinical parameters and estimated signature activities. A more systematic effort will be necessary to make substantial claims in this respect.

Supplementary Figure 25: Mutational signature analysis of PCAWG samples that have been classified as mismatch repair proficient (MMRP, $n = 2,700$) and deficient (MMRD, $n = 41$) in <https://doi.org/10.1038/s41588-024-01659-0> (see Supplementary Table 4). As in Supplementary Figure 24, we compute the fraction of MMRP and MMRD samples, respectively, where signatures associated with defective DNA mismatch repair (SBS6, SBS14, SBS15, SBS20, SBS21, SBS26, and SBS44) are active (left panel) and their mean joint activity in those samples (right panel).

The github is very dense, but everything looks ok and reproducible.

We followed this suggestion and restructured the GitHub code.

Reviewer #3

We thank the reviewer for a positive overall evaluation of the revised manuscript.